# Exploiting weakly supervised visual patterns to learn from partial annotations

**Kaustav Kundu,** **\*Erhan Bas, Michael Lam, Hao Chen, Davide Modolo, Joseph Tighe**
Amazon Web Services
{kaustavk,erhanbas,michlam,hxen,dmodolo,tighej}@amazon.com

## Abstract

As classifications datasets progressively get larger in terms of label space and number of examples, annotating them with all labels becomes non-trivial and expensive task. For example, annotating the entire OpenImage test set [32] can cost \$6.5M. Hence, in current large-scale benchmarks such as OpenImages [32] and LVIS [16], less than 1% of the labels are annotated across all images. Standard classification models are trained in a manner where these un-annotated labels are ignored. Ignoring these un-annotated labels result in loss of supervisory signal which reduces the performance of the classification models. Instead, in this paper, we exploit relationships among images and labels to derive more supervisory signal from the un-annotated labels. We study the effectiveness of our approach across several multi-label computer vision benchmarks, such as CIFAR100 [31], MS-COCO panoptic segmentation [27], OpenImage [32] and LVIS [16] datasets. Our approach can outperform baselines by a margin of **2-10**% across all the datasets on mean average precision (mAP) and mean F1 metrics.

## 1 Introduction

In the past decade, deep networks have shown promising results on a wide variety of applications and tasks, such as image classification [19, 30, 52], image segmentation [17, 18, 40, 43, 50], object detection [9, 14, 18, 49] and vision language tasks, *e.g.* image captioning [26, 60, 65], visual question answering [1, 2, 15, 37, 51], *etc*. As the number of applications increase, the number of visual concepts that are needed to be recognized have also increased. The models need to reason across a wide breadth as well as depth of visual concepts. In Fig. 1, we can see the diversity and fine-graininess of the label categories that can be defined across the images of a dataset. Annotating such large-scale benchmarks is an arduous task [12, 29] since this process needs verification of all possible labels across all images. It is expensive and practically infeasible. For example, using the Amazon Mechanical Turk service to verify all 9.6K labels of the entire OpenImage test set [32] of 116K images (not a particularly large number of images for computer vision tasks) would cost \$6.5M. For specialized domains such as chest radiographs, annotations are ambiguous and must come from high-skilled annotators (aka, doctors in this case) [22, 24].

To overcome this challenge, large scale datasets such as OpenImages [32] and LVIS [16] have adopted the concept of federated datasets [38], where a single dataset is formed by the union of a large number of smaller constituent datasets for a single category. For each category, $c$, there exists two disjoint subsets of the entire dataset, $\mathcal{P}_c$ and $\mathcal{N}_c$, which contain the positive and negative images associated with the class label, $c$ respectively. For OpenImages and LVIS, the sum of positive and negative images per category (on average) is 2-3 orders of magnitude less than the total number of images. Such a process can not only reduce the annotation cost, but also speed up the effort of adding new labels to the dataset. This strategy has been used to fast track annotation of COVID-19 cases [10]

| | | | |
|---|---|---|---|
| Images | 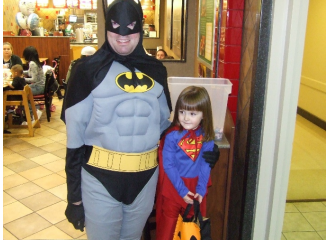 |  |  |
| Pos | Man, Human head, Human arm, Human hair, Clothing, Girl, Human mouth, Cosplay, Woman, Person, Costume, Human face, Superhero, Boy | Real tennis | Real tennis, Racketlon, Tennis skirt, Person, Woman, Sports equipment, Clothing, Ball, Strings, Tennis Equipment, Tennis player, Tennis, Soft tennis, Racquet sport, Human face, Racket |
| Neg | Comics, Convention, Anime | Tennis polo | Hockey, Girl, Rackets, Soccer, Tennis polo, Statue, Basketball (Sport), Elbow, Baseball, Bowling |
| | (a) | (b) | (c) |

Figure 1: Examples of images from OpenImages v6 dataset

alongside existing X-ray imaging datasets such as [24, 62]. In Fig. 1, we show some examples of images and their corresponding positive and negative labels from the OpenImages [32] dataset. It can been seen that similar images can have a large variation in the labels that are annotated. For example, `human head` is present as a positive label in Fig. 1 (a), but not in (b) or (c). While this design principle of enables the computer vision industry and the research community to construct and evaluate on large-scale benchmarks efficiently, it also forces the community to reconsider how to train deep networks on such datasets. Traditional methods to train the classification models on partially annotated datasets simply ignore the un-annotated labels during training but that leaves valuable training signal unused.

In this paper, we explore relationships among the images and labels to derive supervisory signal from the un-annotated label categories. For example, in both Fig. 1 (b) and (c), there is a person with a tennis racket. If the visual representations of these images are close by, we can use the annotations from each other. Thus the `person` label in Fig. 1 (c) can be considered as a positive label for the image in Fig. 1 (b). We can also exploit label relationships in a similar fashion. In Fig. 1 (c), we have `person` as a positive label, but not `human arm` or `human head`. If the representation of `human arm` and `human head` is close to that of the `person`, we can consider them as positive labels as well. Such relationships have been explored in the domain adapation [33] and few shot learning literature [48, 59], where such relationships have been reasoned across multiple datasets, but here we reason about these relationships within the same dataset.

Recently, [13] introduced the problem of training deep neural networks on partially annotated datasets. It works by first training a classification network while ignoring the un-annotated labels. Then, it uses a graph convolutional network to bootstrap on the un-annotated labels and hence expand the annotated labels. In this paper, we introduce a simple, yet effective baseline, where all un-annotated labels are considered as negative. Similar approaches have been attempted for noisy label classification, where the number of un-annotated labels or incorrect labels are significantly less compared to the partial label scenario. While this may not be intuitive for the partial labeling scenario, we observe that this baseline with a simple class-weighting scheme can sometimes out-perform or perform similar to the training approaches where the un-annotated labels are ignored [13, 21]. We hypothesize that this might be because the number of positive labels are orders of magnitude smaller than the negative labels [35, 32, 16]. Hence, this noisy, negative supervisory signal results in learning better representations compared to the scenario where we use a clean, but significantly less supervision signal. While this baseline improves classifier performance by exposing the model to significantly more data, the assumption that all un-annotated labels are negative is clearly incorrect. This form of noise cannot be modeled trivially since it depends on complex factors involving annotation budget and ad hoc choices involved during the annotation process. To overcome this deficiency, we exploit image and label relationships to soften the noise from our new baseline. Instead of ignoring the un-annotated labels or assigning them as strict `negative` labels, we use image and label relationships to consider the un-annotated labels in a "soft" manner. We model this softness using knowledge distillation [20]. We evaluate the performance of our multi-label classification models on challenging datasets such as CIFAR100 [31], MS COCO detection [35], MS COCO panoptic segmentation [27],

OpenImage [32] and LVIS [16]. Our code will be made open source upon acceptance. In summary our contributions are:

- We introduce a simple baseline that outperforms previous methods on the task of training from partially labeled datasets.
- We present a novel method based on image and label relationships to soften the strict negative supervisory signal imposed by our baseline.
- We show the effectiveness of both our baseline and novel approach on 5 challenging, large-scale vision datasets.
- We release all code to reproduce the results presented in this paper.

## 2 Related Work

**Partial labels as noisy labels.** There is a lot of approaches in the literature which deal with the un-annotated labels as noise. Noise can be considered as a statistical outlier which can be modeled to reason about the true probability distribution of the data [8, 36, 39, 41, 53, 57, 58, 64]. There have been different approaches which have the noise function either as a prior [41, 45, 47] or as a function of the label space [53] or the input space [36, 39, 58] or both [8, 57]. These approaches reduce the influence of noise in the data and enable the models to train with the true underlying probability distribution of the data. However, in the case of partial labels, our hypothesis is that modeling of the phenomena which chooses which subset of labels are to be annotated for each image is complex and non-trivial.

Another common approach to deal with the noisy annotations is to bootstrap [7, 45, 47, 67] the labels in an iterative fashion. Bootstrapping on image classification tasks can be done based on either classification scores [23, 47] or clustering based objectives [7, 67]. The former set of approaches uses the top-scoring positives and negatives (from the previous rounds of training) to expand or clean up the label annotations during training the subsequent training rounds. The latter set of approaches determine pseudo-labels with the clustering objective and use them to train a classification model with a cross-entropy loss function. Our approach is significantly different from this line of work. Treating un-annotated labels as noisy negative labels or statistical outliers indicate that the noise is considerably less than the training signal. However, in the case of partial labels, such assumptions cannot be made. Moreover, bootstrapping is an iterative approach in which label expansion occurs. While, ours is a single round of training which incorporates information from the un-annotated labels in a soft manner. Training in an iterative fashion can potentially help our performance further, but that's beyond the scope of this paper.

**Positive Unlabeled (PU).** A special case of partial label annotations the case of positive unlabeled or missing label problem [4, 28, 42, 69]. In this case, the samples are annotated with a set of positive labels, and the remaining un-annotated labels are either positive or negative. There is no explicit representation of annotated negative set. In our approach, we consider the more general and realistic setting, where both the annotated and the un-annotated labels have both positive and negative classes. In these approaches, it is common to either use the un-annotated labels as ignore or strict negatives.

**Partial labels.** Approaches such as [5, 25, 56, 63, 66, 68, 70] have proposed different optimization strategies to deal with the partial labeling dataset. However, such approaches are not scalable to large scale datasets. Moreover, end-to-end training of a deep network along with these optimization strategies is not trivial. Approaches have mostly focused on using the un-annotated classes as ignore classes [13]. Biomedical classification approaches [3, 22] have considered the possibility of using the un-annotated labels as positive or negative. However, this has not been used to baseline computer vision benchmarks.

[13] proposes an iterative algorithm which uses a graph convolutional network to expand the label space and "annotate" the un-annotated labels. In this work, we aim to improve single iteration of training. We can use similar approaches to iteratively improve the performance. Moreover, the results demonstrated in their work do not bridge the gap to the oracle approaches as much as our single round of training. [46] suggests an approach where they explicitly make the assumption about how the individual datasets have clean and complete annotations, but it is not complete across the datasets. Theirs is a special case of our approach. Our approach can handle completeness and incompleteness

in each individual dataset as well as across multiple datasets. We show results across all these settings in our experimental section.

**Soft Labeling.** There have been several notable works [54, 55, 71] in the literature which treat the classes as soft labels and exploit the label relationships to enable better visual representation for each visual class category. Recent approaches [44] have used teacher-student networks to determine a target distribution of classes, which is a softer version compared to the usual one-hot encoding of image classification training settings. The approach adopted by [34] is similar to the approach we have adopted here. However, they have adopted a constant temperature term for label smoothing while adopting a softer version of the target distribution from a network trained on clean dataset. However, in our case, we don't have access to a "clean", fully annotated dataset.

## 3 Approach

Given an input image, $\mathbf{x}_i$ and a set of visual categories, $\mathcal{C}$ ($|\mathcal{C}| = C$), we are interested in finding the visual concepts among $\mathcal{C}$ that are present in this image. The GT annotations provided for this image, are a set of positive labels, $\mathcal{P}(\mathbf{x}_i) \subset \mathcal{C}$ and a set of negative labels, $\mathcal{N}(\mathbf{x}_i) \subset \mathcal{C}$, where, $\mathcal{P}(\mathbf{x}_i) \cap \mathcal{N}(\mathbf{x}_i) = \emptyset$ and typically $| \mathcal{P}(\mathbf{x}_i) \cup \mathcal{N}(\mathbf{x}_i) | \ll | \mathcal{C} |$. Let $\theta$ denote the parameters in the deep network, $f(.; \theta)$, which takes as input an image and outputs a $C$-dimensional output $\in \mathcal{R}^C$. The prediction output of the network, $y_{i,c}$, for category, $c$, for an input, $\mathbf{x}_i$, is given by $y_{i,c} = \sigma(f_c(\mathbf{x}_i; \theta))$ where $\sigma(a) = 1/[1 + \exp(-a)]$. $N$ is the number of training examples, and since we consider the training examples to be i.i.d, we take the overall loss function to be averaged across all training examples in a batch. For the purpose of brevity, we ignore the notation, $i$ to denote the training example from hereon.

We define different settings in which the BCE loss function can be applied in the partial labeling multi-label image classification problem.

**No exposure** (*NE*): In this setting, we train only on the labels that have been annotated for the input image, $\mathbf{x}$. Thus the GT labels are not "exposed" to the un-annotated labels. The loss function for this setting is as follows:

$$L_{\text{BCE}}^{\text{NE}}(\mathbf{x}, \mathbf{y}) = - \sum_{c^+ \in \mathcal{P}(\mathbf{x})} \log \sigma(y_{c^+}) - \sum_{c^- \in \mathcal{N}(\mathbf{x})} \log \sigma(-y_{c^-}) \tag{1}$$

**Weighted no exposure** (*wNE*): Following [13], we also consider a weighted no exposure (*wNE*) setting, where the loss term of each sample is weighted by a factor which is a function of the number of annotated labels corresponding to that input image.

**Full exposure** (*FE*): In this setting, we "expose" the set of all labels, not just the labels that have been annotated, during training for the input image.. Unless mentioned, we treat the GT value of the un-annotated labels as negative. While this may seem counter intuitive, in the majority of cases this will be the correct label as the number of negative labels outnumber the number of positive labels by orders of magnitude. However, in some cases [22], assigning them have positive as given better performance. If we represent the set of un-annotated labels as $\mathcal{U}(\mathbf{x}) = \mathcal{C} - (\mathcal{P}(\mathbf{x}) \cup \mathcal{N}(\mathbf{x}))$, the loss function can be written as follows:

$$L_{\text{BCE}}^{\text{FE}}(\mathbf{x}, \mathbf{y}) = - \sum_{c^+ \in \mathcal{P}(\mathbf{x})} \log \sigma(y_{c^+}) - \sum_{c^- \in \mathcal{N}(\mathbf{x})} \log \sigma(-y_{c^-}) - \sum_{c^u \in \mathcal{U}(\mathbf{x})} \log \sigma(-y_{c^u}) \tag{2}$$

Using this loss function out of the box does not usually perform well. But assigning a higher weight for the positive classes enables this simple strategy to outperform previously proposed method of [13] (wNE). But it can suffer especially for rare classes where labeling a few positives as negatives can completely overwhelm the labeled positives.

**Soft exposure** (*SE*): Both no exposure and full exposure methods are simple but have vital flaws. The full exposure negative setting can overwhelm the network while the no exposure setting will miss out on a large number of negative classes, which are not seen during training. These two issues are complementary to each other. Hence we propose a soft exposure loss function, where instead of treating the labels as hard negatives as done in the **FE** setting, we use label smoothing to reduce its training signal while not completely ignoring it as in the **NE** setting does. Hence, the loss function in

this setting is as follows:

$$L_{\text{BCE}}^{\text{FE}}(\mathbf{x}, \mathbf{y}) = - \sum_{c^+ \in \mathcal{P}(\mathbf{x})} \log \sigma\left(y_{c^+}\right) - \sum_{c^- \in \mathcal{N}(\mathbf{x})} \log \sigma\left(-y_{c^-}\right) - \sum_{c^u \in \mathcal{U}(\mathbf{x})} \log \sigma\left(y_{c^u}, T\right) \quad (3)$$

where, $\sigma\left(a, T\right) = 1/[1 + \exp(-a/T)]$. The idea of temperature is to not consider the un-annotated negative labels as hard negatives, but as soft labels (positive/negative). The BCE loss function forces the predictions to be more confident to get a lower loss value. Lowering the temperature value, converts the prediction value to be more confident and hence have a lesser loss value without changing the actual prediction from the model. This can be useful for the un-annotated labels, which gets a softer loss value compared to the hard labels in the full exposure model. At the same time, it provides negative information which is not provided in the no exposure model.

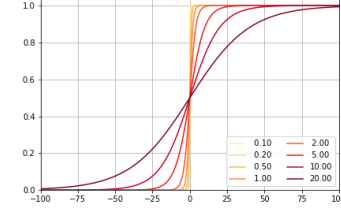

Figure 2: Sigmoid values for different temperature settings

Each un-annotated labels is assigned a unique temperature values based on how confident we are in its hard label. We use the distance between an un-annotated label and the annotated labels to model these temperature values. The closer the un-annotated labels are to the annotated labels, more confident are we with the positive or negative class assigned to the un-annotated label. To measure the close-ness between an un-annotated label and the annotated labels, we use two types of similarity functions: one based on label relationships, $(d_L)$ and one base on image relationships, $(d_I)$.

Let the distance between the two label categories, $c_1, c_2$, be denoted by $d\left(c_1, c_2\right)$, where $d$ can be Euclidean or cosine distance functions. The feature vector associated with the label category, $c$ (denoted by $\psi\left(c\right)$) represents the median of all image representations, where $c$ is present. The distance between an un-annotated label, $c^u$ and the positive annotated labels, $\mathcal{P}\left(\mathbf{x}\right)$, is computed as follows:

$$d_L^+\left(c^u, \mathbf{x}\right) = \min_{c_i \in \mathcal{P}(\mathbf{x})} d\left(c, c_i\right) \quad (4)$$

Similarly, we can compute the distance, $d_L^-\left(c^u, \mathbf{x}\right)$, between an un-annotated label, $c^u$ and the negative annotated labels, $\mathcal{N}\left(\mathbf{x}\right)$.

For image-relationships, we consider the positive and negative labels of the images which are nearest neighbors of the current image. Let, $NN_k\left(\mathbf{x}\right)$ represent the $k-$nearest neighbors of the input sample, $\mathbf{x}$. For each un-annotated label, $c^u \in \mathcal{U}\left(x\right)$, we consider all the images in $NN_k\left(\mathbf{x}\right)$ where, $c^u$ is annotated as positive as, $\mathcal{P}^I\left(c^u, \mathbf{x}\right) = \{\mathbf{x}_i \mid \mathbf{x} \in NN_k\left(\mathbf{x}\right) \cap c^u \in \mathcal{P}\left(\mathbf{x}_i\right)\}$. Similarly, we can define, all the images in $NN_k\left(\mathbf{x}\right)$ where, $c^u$ is annotated as negative as, $\mathcal{N}^I\left(c^u, \mathbf{x}\right) = \{\mathbf{x}_i \mid \mathbf{x} \in NN_k\left(\mathbf{x}\right) \cap c^u \in \mathcal{N}\left(\mathbf{x}_i\right)\}$. We can define, the distance of an un-annotated label, $c^u$ in w.r.t. a hard positive label, as follows:

$$d_I^+\left(c^u, \mathbf{x}\right) = \begin{cases} 1, & \mathcal{P}^I\left(c^u, \mathbf{x}\right) = \phi \\ \min_{\mathbf{x}_i \in \mathcal{P}^I(c^u, \mathbf{x})} d\left(\mathbf{x}, \mathbf{x}_i\right), & \text{otherwise} \end{cases} \quad (5)$$

Similarly, we can define the distance of $c^u$ w.r.t. a hard negative label $(d_I^-\left(c^u\right))$. We can combine the $d_L^+\left(c^u, \mathbf{x}\right), d_I^+\left(c^u\right), d_L^-\left(c^u, \mathbf{x}\right), d_I^-\left(c^u, \mathbf{x}\right)$, as follows:

$$d^+\left(c^u, \mathbf{x}\right) = \min\left(d_L^+\left(c^u, \mathbf{x}\right), d_I^+\left(c^u, \mathbf{x}\right)\right) \quad (6)$$

$$d^-\left(c^u, \mathbf{x}\right) = \min\left(d_L^-\left(c^u, \mathbf{x}\right), d_I^-\left(c^u, \mathbf{x}\right)\right) \quad (7)$$

Finally, the temperature is set as follows:

$$T\left(c^u, \mathbf{x}\right) = \begin{cases} \exp\left[\beta\left(1 - d^+\left(c^u, \mathbf{x}\right)\right)\right] + \gamma, & d^+\left(c^u, \mathbf{x}\right) < d^-\left(c^u, \mathbf{x}\right) \\ -\exp\left[\beta\left(1 - d^-\left(c^u, \mathbf{x}\right)\right)\right] + \gamma, & \text{otherwise} \end{cases} \quad (8)$$

Since distance values are non-negative and can be arbitrarily large, we found that normalizing the distance values significantly helps in the performance. Each of $d^+\left(c^u, \mathbf{x}\right), d^-\left(c^u, \mathbf{x}\right)$ is normalized by the maximum value of $d^+, d^-$ among all un-annotated labels for the sample, $\mathbf{x}$. In our experiments, we use, $k = 10$, $\beta = 5.0$ and $\gamma = 0$. For computing the distance between labels and finding the nearest image neighbors, we use representations from a pre-trained Imagenet network.

# 4 Results

We analyze the performance of our approach and the baselines on challenging multi-label datasets from the computer vision literature. To evaluate the approaches on these benchmarks, we use mean AP and mean F1 (averaged across classes) metrics. For the F1 score, the class-based score thresholds are set based on the best F1 score on the validation set.

## 4.1 Synthetically generated partially annotated datasets

To understand how our model performs on the partially annotated datasets, we use (almost) fully annotated datasets to compare proposed models against oracle models at different fractions of partial label annotations. We use CIFAR100 [31], MS COCO detection [35] and MS COCO panoptic segmentation [27] datasets for the synthetically generated partially annotated datasets. For generating the synthetic partially annotated datasets, we follow two strategies, (a) random removal of labels from the fully annotated datasets as done in [13], and (b) knowledge graph based partially annotated dataset generation. In short, our knowledge graph strategy takes inspiration from the way many datasets such as [11, 35], are labeled based on a hierarchy of visual concepts. Here, during the verification of labels in an image, the process starts with verifying visual concepts that are present in the first level of the hierarchy. If a visual concept is present in the image, then the verification of lower levels of hierarchy is done to explore more fine-grained visual concepts. For our knowledge graph strategy we use a similar process to generate partial label annotations. Instead of fully annotating an image all the way down to leaf nodes, we randomly choose a node in the knowledge graph. All labels from the root until that node are considered as verified positive annotations. The sub-tree below the node is considered as un-annotated labels.

### 4.1.1 Randomly generated partially annotated datasets

We use the MS-COCO 2017 detection benchmark [35] for this purpose. It has 80 categories among 118K training images 5K test images. We randomly split the original training set into 116K and 2K validation images, *s.t.* the training and validation sets have same label distribution.

**Synthetic partial dataset creation.** MS COCO detection dataset is known to have $\sim 85\%$ recall in the positive label annotations [35]. But for purposes of the experiments in this section, we consider them to be fully annotated. We create subsets of MS COCO detection dataset with $5\%, 10\%, 30\%, 50\%, 70\%, 90\%$ of the positive labels in the dataset. 100% labels indicate the original annotations of the dataset. In this dataset, there are $\sim 7$ positive labels per image. The remaining $\sim 73$ labels are considered negative. To have similar number of positive and negative labels as the realistic datasets, we choose the negative labels randomly *s.t.* #negative annotations is $1 - 1.5\times$ #positive labels. This approach mirrors the actual partially annotated datasets and is different from the positive-unlabeled scenario in [13], where all negative annotations have been considered. For each fraction of partially annotated datasets, we create 5 separate subsets with different random seeds. The results are averaged across all 5 subsets in this section.

**Experimental details.** We followed the same network (ResNeXt101 pretrained on ImageNet) and optimization strategies as used in [61]. The networks have been trained using the sgd optimizer with an initial learning rate of 0.001, momentum of 0.9 and weight decay of 0.0005. The learning rate is decreased by a factor of 10 at the end of $10^{\text{th}}$ and $20^{\text{th}}$ epochs. We use a batch size of 24 with the input image dimension as $224 \times 224$. The networks are trained for 36 epochs. During testing we choose the model which has the highest mAP score on the validation dataset.

**Results.** The results of the different training settings: NO EXPOSURE (NE), WEIGHTED NO EXPOSURE (wNE) [13], FULL EXPOSURE (FE) and SOFT EXPOSURE (SE) are shown in Fig. 3 (a). We observe that wNE performs slightly better than the NE setting due to better stability in the optimization function. The FE model performs slightly better than wNE, which shows that the lack of negative annotations can impact performance. Because in these computer vision benchmarks naturally have far more negatives, labeling all un-labeled annotations as negative annotations (FE) is not a bad baseline since it is true most of the time. The false positive in the GT can be considered as noise, and its impact is significantly more when the dataset moves into the 10% partially annotated regime. Moreover, the noise in the FE model is alleviated by adding more weight to the positive classes. In our experiments, we use a constant multiplicative weight of 5.0 for positive classes and 1.0 for the negative classes.

In Fig. 3 (b), we show an ablation study of the temperature modeling to capture softness in the hard targets. The LABEL SMOOTHING (LS) model captures the softness when all un-annotated labels

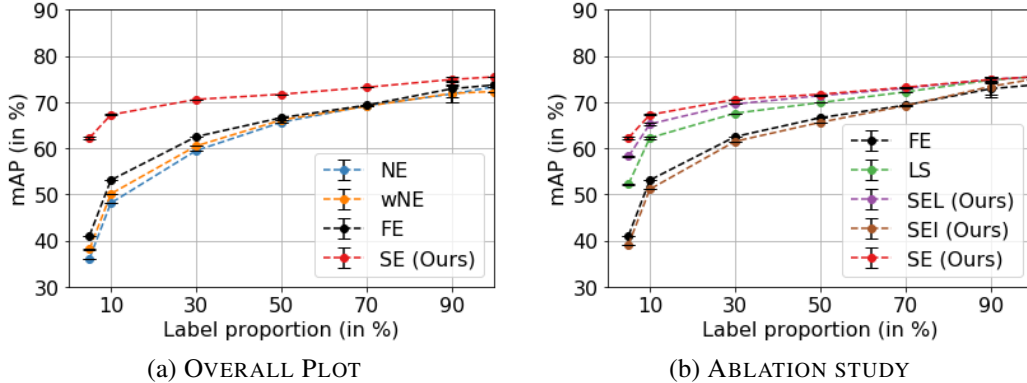

| (a) OVERALL PLOT | (b) ABLATION STUDY |

Figure 3: Test AP vs fraction of partially annotated dataset (MS COCO detection dataset).

have a constant temperature value and is not dependent on any label or image relationships. The SE-L and SE-I model captures only the label and image relationships respectively to determine the degree of softness of the un-annotated labels. The SE-I model by itself does not perform that well, because the labels from the nearby images do not cover all the un-annotated labels and hence performs close to the NE and wNE models. But combined with the SE-L model, it can improve the overall performance since the confidence of un-annotated labels become higher (and hence the temperature becomes higher).

### 4.1.2 Knowledge-graph based partially annotated dataset generation

For this purpose we use the MS COCO panoptic segmentation [27] as they have more label categories and enable us to analyze this section better. It includes 80 thing categories and 53 stuff categories. We use the knowledge graph defined in [6] and merge it with the WordNet. This results in a total of 163 label cateogories. We do not create different fractions of partial label annotations as in the previous section. At lower fractions, the leaf labels would be missing completely and the visual categories belonging only in the upper levels of the knowledge graph would be present. That would prevent us from analyzing the performance across all labels in the dataset across all fractions of labels. Instead, we simply create 5 different datasets based on different random seeds and show results averaged among them. The knowledge graph and the details of the dataset creation is described in the supplementary paper.

**Experimental details.** We use the same training details as done for the MS COCO detection benchmark, mentioned in Sec. 4.1.1.

**Results.** The overall results are shown in Tab. 1. The performance of the SE based models are significantly better than the baselines. The oracle model has access to all annotations. There are two reasons which contribute to why the performance of the SE based models are better than the oracle model. Firstly, label smoothing tends to improve the performance of the labels. Secondly, since MS COCO panoptic segmentation dataset is known to have missing labels, using the image and label relationships can improve the performance of classification models over the oracle model.

We plot the convergence of the validation AP metric vs epochs is shown in Fig. 4. SOFT EXPOSURE (SE) model converges faster compared to the baselines due to additional supervisory signal from the un-annotated labels. We use one of the random dataset to plot the labelwise performance in Fig. 5 and analyze them in terms of the fraction of labels annotated in the dataset. On the y-axis, we plot the label-wise difference of the AP metric of SE model w.r.t. the best performing baseline (wNE) model. Positive values indicate the labels where the SE model performs better than the wNE model. The labels are sorted from the worst to the best performing ones. The brighter bars indicate higher fraction of labels being annotated, while the darker bars indicate labels where most of them are un-annotated. We observe that most of the darker bars are closer to the right part of the figure and the brighter bars are closer to the left part of the figure. This indicates that our approach does better for labels which higher fraction of labels are un-annotated.

### 4.2 Real partially annotated datasets

For studying the performance on realistic partially annotated datasets, we use OpenImage [32] and LVIS [16] datasets. Please refer to the supplementary material for the results on the LVIS dataset. In

| Loss Setting | Validation | | Test | |
|---|---|---|---|---|
| | mAP | mF1 | mAP | mF1 |
| Oracle | 0.6442 | 0.6353 | 0.6279 | 0.6082 |
| NE | 0.5943 | 0.5978 | 0.5770 | 0.5648 |
| wNE [13] | 0.6148 | 0.6079 | 0.5925 | 0.5675 |
| FE | 0.6127 | 0.6071 | 0.6012 | 0.5641 |
| LS | 0.6418 | 0.6281 | 0.6347 | 0.5877 |
| SE-I (Ours) | 0.6487 | 0.6331 | 0.6371 | 0.5815 |
| SE-L (Ours) | 0.6544 | 0.6378 | 0.6456 | **0.6070** |
| SE (Ours) | **0.6578** | **0.6403** | **0.6491** | 0.6048 |

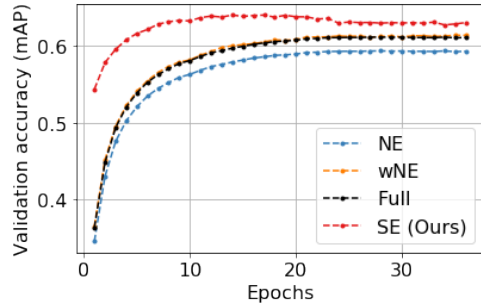

Table 1: **MS COCO PS results**. **Bold**: best performing and Underlined: 2nd best.

Figure 4: **Faster covergence of SE**: Validation mAP vs epochs

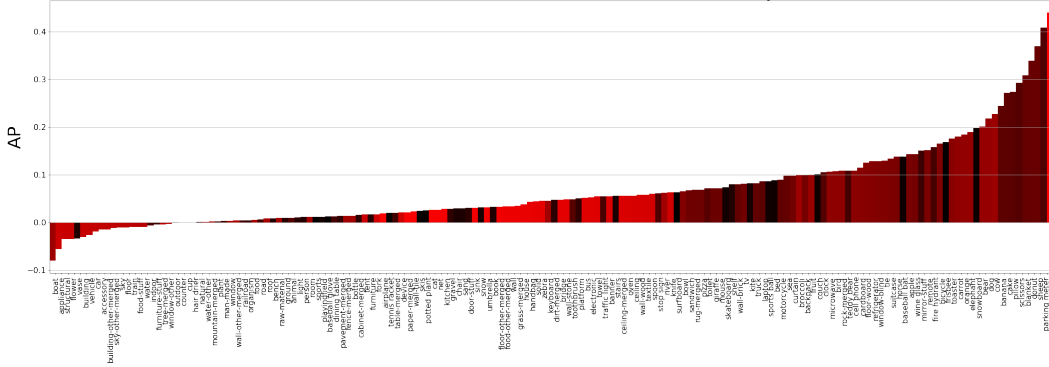

Figure 5: MS COCO Label-wise Performance (Sorted by the AP score). Increasing color brightness of the bars indicate increasing **fraction of missing annotations**. Best viewed in color

this paper, we use the detection dataset of OpenImagev5 and convert it into a multi-label classification benchmark. It has 1.6M training, 33K validation and 113K test images with 497 label categories.

**Experimental Details.** We choose the ResNet-101 architecture, pre-trained on ImageNet dataset. As in pre-trained ImageNet dataset, we resize the image to $224 \times 224$ as input to the network. We used a batch size of 96 per GPU and random vertical and hortizontal flips for data augmentation. An initial learning rate for all layers except the last was set at $2\times10^{-4}$. The initial learning rate for the last layer was set at $2\times10^{-5}$. We used the Adam optimizer with learning rate decay of 0.3 at the end of 8th and 15th epochs. The model was trained for 30 epochs. All the models have been trained with these hyper-parameter settings.

**Results.** We report the performance of the models on the test set of OpenImages detection benchmark in Tab. 2. Since the all the labels for the test images are not annotated, we only evaluate the performance of our model on the set of annotated labels. Hence false positive can happen only if a `positively annotated` label is predicted as a negative class. Similarly, false negative can happen only if a `negatively annotated` label is predicted as a positive class. We observe that our approach performs significantly better compared to the baseline models by a margin of $\sim$3%.

## 5 Conclusion

In this paper, we look at the how we can use modern day datasets which are annotated with partial labels for the task of multi-label image classification. We argue that instead of looking at partial labeling datasets as noisy labeled dataset, we should adopt our training strategy in a manner which can use the un-annotated labels to the benefit of training. In that direction, we exploit image and label relationships to get more supervisory signal from the un-annotated labels. We hope this can be fuel interests in the community to further explore the partial label problem for a wide range of applications and tasks.

| Training Strategy | NE | wNE [13] | FE | LS | SE-I (Ours) | SE-L (Ours) | SE (Ours) |
|---|---|---|---|---|---|---|---|
| mAP | 75.61 | 77.18 | 76.89 | 79.32 | 77.56 | 80.17 | **80.67** |

Table 2: OpenImage results

# 6 Broader Impact

We investigate the issue of training classification models in partially annotated datasets. Partially annotated datasets can enable democratization of AI and allow small and large organizations to construct and train models with the requirement of large data sources. The partially annotated datasets are realistic in nature. We hope that in-depth studying of this problem can start more formal discussions into fairness issues concerning large-scale datasets.

# 7 Funding Sources

This work has been entirely done at Amazon.

# 8 Acknowledgment

We would like to thank Meng Wang for the initial push towards thinking in this direction; Aditya Deshpande for his insights into approaching this problem and encouragement; Howard Wu for his help during discussions; Rahul Bhotika for his support.

# 9 Highlights of the Rebuttal based on Reviewer Feedback

PRIOR WORK. The state of the art approach for the partially annotated multi-label classification task is [13]. There are two contributions of [13]. One of them is the nWE baseline, which our approach outperforms. The other contribution is using GNN. But it barely has any improvement. In our settings we do see a similar trend of achieving $< 0.2\%$ improvement in mAP. From a high level, instead of modeling label relations directly from the data, we use priors in terms of distance of class embeddings. Moreover, we exploit image similarities as well in this approach.

SENSITIVITY TO HYPER-PARAMETERS. We selected the values of $\beta = 5$ and $\gamma = 0$ based on the validation set. The mAP is within 1% of the reported performance (on average) for $\gamma \in (0, 0.1]$. The drop in performance can be as large as 5% with $\gamma \in (0, 1]$. For, $\beta \in \{1, 2, 5, 10, 20, 50, 100\}$, avg. mAP on the validation set was within 2% of the best performance at $\beta = 5$. For values of $5 < k \leq 30$, the SEI performance increases by 5%, but it's improvement on the SE model is $< 0.15\%$. $k < 5$, reduces the performance of SEI and SE models and brings them closer to NE and SEL models respectively.

DESIGN CHOICES OF SE MODELING. The main motivation of the paper is to use image-image and label-label relationships to capture more supervisory signal from the unsupervised un-annotated labels. We implemented this via temperature modeling. Exploring better modeling choices is a work in progress. Thank you for suggesting the entropy based modeling. Regarding the "hard" minimum operations, we also experimented with "softer" operations in this rebuttal. Instead of taking the minimum, we take the median operation in Eq. (4) and Eq. (5) among the top 5 neighbors. We see an improvement of $\sim 0.7\%$ for label-label relationships using this approach. While both label and image based relationships improve the performance, we do observe that the label based distances dominate the image based. This is because the number of labels considered for the image based distances is significantly lesser than the label based distances. While additional 72.7 labels are considered for the label based, the number of labels being considered for image based is $\sim 5.5$ @10% partially annotated data. We will perform an in-depth analysis of the effect of this discrepancy on our approach.

DISTANCE COMPUTATION COST. It takes <1 epoch training time ( $\sim 15$ min. on a single V100 GPU) and it's done once.

FEATURE PRE-PROCESSING FOR DISTANCE COMPUTATIONS. We process the features in the same way as [67], where we use the 2048-dim feature vector and do L2 normalization on them. For k-NN, we use these features to compute the neighbors. For $\psi(c)$, we take a median of these representations across all images where, $c$ occurs. We had also experimented with mean, but found median to have better performance.

$d_L$ DISTANCES WHEN $P(x), N(x) = \phi$. Implementation-wise, we ignore such labels when this happens. However, when combined with dI , the overall distance value defaults to 1 based on Eq. (5).

## Footnotes

*Primary correspondence

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
