[Supplementary Material]

# Suppl. Material: Exploiting weakly supervised visual patterns to learn from partial annotations

## 1 Additional Results

### 1.1 Synthetically generated partially annotated datasets

#### 1.1.1 Knowledge-graph based partially annotated dataset generation

The knowledge graph used for the MS COCO panoptic segmentation dataset results [7] in Appendix B.

### 1.2 Real partially annotated datasets

In this section, we discuss results on the LVIS dataset [6]. LVIS dataset has 57K training and 5K test images with 1200 categories. The label categories are categorized into three categories based on their frequencies. We use the two highest frequency ones which result in 776 label categories. We use 3332 images from the training set for our validation set while maintaining the same label distribution.

**Experimental details**. We followed the same network (ResNeXt101 pretrained on ImageNet) and optimization strategies as used in [14]. The networks have been trained using the sgd optimizer with an initial learning rate of 0.001, momentum of 0.9 and weight decay of 0.0005. The learning rate is decreased by a factor of 10 at the end of $10^{th}$ and $20^{th}$ epochs. We use a batch size of 24 with the input image dimension as $224 \times 224$. The networks are trained for 36 epochs. During testing we choose the model which has the highest mAP score on the validation dataset.

**Results**. We report the performance of the models on the test set of the LVIS benchmark in Tab. 1. Since the all the labels for the test images are not annotated, we only evaluate the performance of our model on the set of annotated labels. Hence false positive can happen only if a `positively annotated` label is predicted as a negative class. Similarly, false negative can happen only if a `negatively annotated` label is predicted as a positive class. We observe that our approach performs significantly better compared to the baseline models by a margin of $\sim$2%.

### 1.3 Partial label annotations when training models across multiple datasets

One of the common scenarios where datasets are partially annotated is when we have multiple datasets. Let us consider two datasets as shown in Fig. 1. We observe the following sources of partial label annotations.

1. *Missing instance*: This problem occurs when a thing/stuff is visually present in the image, but any visual concept related to that thing/stuff was not defined while that dataset was annotated. Let's say that the label TENNIS is present in both datasets, while PERSON, TENNIS RACKET is present only in $\mathcal{D}_2$. If the image in Fig. 1 (b) is present in $\mathcal{D}_1$, it will be labeled as tennis. However, if the image in Fig. 1 (c) is present in $\mathcal{D}_2$, it will be labeled as TENNIS, PERSON, TENNIS RACKET. In this case, while each of the latter categories are present visually in $\mathcal{D}_1$, they can be considered as negative during training. However, they would be

| Training Strategy | NE | wNE [3] | FE | LS | SE-I (Ours) | SE-L (Ours) | SE (Ours) |
|---|---|---|---|---|---|---|---|
| mAP | 19.14 | 22.45 | 22.18 | 22.56 | 22.51 | 23.76 | **24.11** |

Table 1: LVIS results

| | | | |
|---|---|---|---|
| **Images** | | | |
| **Pos** | Man, Human head, Human arm, Human hair, Clothing, Girl, Human mouth, Cosplay, Woman, Person, Costume, Human face, Superhero, Boy | Real tennis | Real tennis, Racketlon, Tennis skirt, Person, Woman, Sports equipment, Clothing, Ball, Strings, Tennis Equipment, Tennis player, Tennis, Soft tennis, Racquet sport, Human face, Racket |
| **Neg** | Comics, Convention, Anime | Tennis polo | Hockey, Girl, Rackets, Soccer, Tennis polo, Statue, Basketball (Sport), Elbow, Baseball, Bowling |
| | (a) | (b) | (c) |

Figure 1: Types of partial label annotations

considered as positives in $\mathcal{D}_2$. This discrepancy could result in sub-optimal performance, especially when this is a frequently occurring phenomena.

2. *Fine-grained mismatch problem*: This problem occurs when a parent label (*e.g.* person) is present in both $\mathcal{D}_1$ and $\mathcal{D}_2$ datasets, but the child label(s) (*e.g.* MAN, GIRL) is present only in one of the datasets, say $\mathcal{D}_2$. For example, if the image in Fig. 1 (a) is present only in $\mathcal{D}_1$, it would be labeled as PERSON. But if the image in Fig. 1 (b) is present in $\mathcal{D}_2$, it would be labeled as PERSON, MAN. This discrepancy in labeling could affect the embedding space to get confused whether labels like PERSON, MAN, GIRL are related to the same visual concept or not.

Thus even if the datasets are fully and correctly annotated, partial labels can still occur while training across multiple datasets. Learning paradigms such as *lifelong learning, continual learning, incremental learning* have been developed to keep training a model on increasing label sets. However, incremental learning approaches suffer from the catastrophic forgetting problem [4, 5, 11, 12, 13]. Even current state of the art approaches have a forgetting rate of $10 - 15\%$. We study the partial annotation problem here and use the baselines described in the paper and our training approach to analyze this problem from the multi-dataset training perspective. One of the key differences of this approach compared to incremental learning approaches is that we do use all the images across all datasets, which is more expensive in terms of memory used to train our network. We do not propose this approach as an incremental learning approach, but provide a new "oracle" baseline for the incremental learning approaches.

We use the CIFAR100 [8] and MS COCO panoptic segmentation [7] datasets for this purpose.

### 1.3.1 Multi-label CIFAR dataset

We used the CIFAR-100 dataset [8] for this purpose. There are 20 super-classes, each of which have 5 children, forming a total of 100 classes. We added more labels to this structure to replicate a similar hierarchy structure such as RKGv2. The root of the tree sub-divides into two children, NATURAL and THINGS. Their sub-trees are shown in Fig. 9 in Appendix A. We defined the subsets in the manner as shown in Tab. 2. The common classes to both datasets are LARGE_MAN-MADE_OUTDOOR_THINGS and PEOPLE and its leaf classes, *i.e.*, images of these classes are labeled just as is for both datasets. Second row has the images belonging to the sub-tree corresponding to the left class being labeled as the right-class for the Dataset 1. The third row has the similar thing, but for Dataset 2. Roughly

| Common super-classes | LARGE_MAN-MADE_OUTDOOR_THINGS, PEOPLE |
|---|---|
| Missing super-classes in Dataset 1 | HOUSEHOLD_ELECTRICAL_DEVICES → HOUSEHOLD_ITEMS<br>VEHICLES_1 → VEHICLES<br>VEHICLES_2 → VEHICLES |
| Missing super-classes in Dataset 2 | FLOWERS → NATURAL<br>FRUIT_AND_VEGETABLES → NATURAL |

Table 2: Subset class groups for multi-label CIFAR-100 dataset.

| Scenario | Wide ResNet | | | | DenseNet | | | |
|---|---|---|---|---|---|---|---|---|
| | Validation | | Test | | Validation | | Test | |
| | mAP | meanF1 | mAP | meanF1 | mAP | meanF1 | mAP | meanF1 |
| Oracle | 0.7686 | 0.7717 | 0.7638 | 0.7483 | 0.7789 | 0.7739 | 0.7818 | 0.7538 |
| FE | 0.6743 | 0.70006 | 0.6789 | 0.6713 | 0.6618 | 0.7101 | 0.6972 | 0.6847 |
| NE | 0.6551 | 0.6789 | 0.6529 | 0.6515 | 0.6955 | 0.7153 | 0.6982 | 0.6868 |
| wNE | 0.6712 | 0.6885 | 0.6553 | 0.6546 | 0.7074 | 0.71140 | 0.7020 | 0.6827 |
| LS | 0.6896 | 0.7052 | 0.6868 | 0.6748 | 0.7124 | 0.7165 | 0.7113 | 0.6887 |
| SE-I | 0.7135 | 0.7202 | 0.7017 | 0.6878 | 0.7427 | 0.7440 | 0.7332 | 0.7101 |
| SE-L | 0.7536 | 0.7428 | 0.7533 | 0.7114 | 0.7749 | 0.7546 | 0.7851 | 0.7322 |
| SE | **0.7766** | **0.7570** | **0.7729** | **0.7228** | **0.7914** | **0.7690** | **0.7909** | **0.7360** |

Table 3: Mean AP and Mean F1 score on the Validation and Test sets of Multi-label CIFAR-100

Dataset 1 have 3x more data as the Dataset 2, with a total of 45k images across both. The validation and test sets have 5k and 10k image respectively.

**Experimental Details**. We experimented on 40-layer Wide Resnets and DenseNets, with wide factor of 4 and growth rate of 40 respectively. We use the same training schemes as in the original Wide Resnet and DenseNet papers.

**Results**. We show the mean F1 scores on the validation and test sets in Tab. 3.

In Fig. 2, we analyze the performance based on different categories of how the labels are annotated in our CIFAR100 subset datasets. The temperature based model corresponds to our proposed SE approach. The different label categorizations are defined above.

72

(a) F1      (b) AP

Figure 2: Subset Category-wise performance on CIFAR100 test set

Figure 3: MS COCO Label Categorization Statistics

Figure 4: MS COCO Label-wise Image Frequency Statistics

### 1.3.2 MS-COCO panoptic segmentation dataset

In this section, we show results on the MS-COCO panoptic segmentation dataset [10, 7] which includes 80 thing categories and 53 stuff categories. for this purpose. To build the knowledge graph, we start from the one defined in [1]. We add some additional parent labels such as ROOM, MAN-MADE, ORGANISM, DEVICE and NATURAL. The final knowledge graph is shown in Appendix B. The final number of all classes in the dataset is 164.

We divide the classes in a way such that the number of labels in Dataset 1 is 57 and the number of labels in Dataset 2 is 139 while having 32 labels in common. The category specific pie charts which show the categorization of labels are shown in Fig. 3. This results in 46 labels being correctly annotated, *i.e.*, the number of images that contain these labels match the oracle scenario. In this setting, there are 93 labels that suffer from the *missing instance* problem and 23 labels suffer from the *fine-grained mismatch* problem. The label-wise statistics are shown in Fig. 4. The number of training images in the Datasets 1 and 2 is $\sim$ 67K (58%) and $\sim$ 48K (42%) respectively.

(a) F1                  (b) AP

Figure 5: Subset Category-wise performance on MS COCO test set

**Overall Results**. We followed the same network (ResNeXt101 pretrained on ImageNet) and optimization strategies as used in [14, 3]. We plot the mean F1 scores on the validation set as a function of epochs in Fig. 5(a). Using our proposed SE approach improves the performance over naive full-exposure and no-exposure settings (and also the oracle model!).

**Quantitative results on different types of missing label problems**. In Fig. 5, we analyze the performance based on different categories of how the labels are annotated in our MS COCO subset datasets. The labels which are correctly annotated in both subsets (*Correct Annotations*) have the best performance.

**Performance vs fraction of missing labels**. In Fig. 6, we plot the label-wise difference of the AP performance of our approach compared with that of the ORACLE, FULL EXPOSURE and NO EXPOSURE settings. Red represents labels with the missing instance problem, green indicates the labels with the fine-grained mismatch problem and blue represents the labels which are correctly annotated. Within each categorization, the labels are sorted based on the improvement in the performance of our approach. The darkest color coding of the bar represents lesser noise in the label annotation, while the brightest color coding indicates more noise in the label annotations. We observe that as the fraction of noisy annotations increase, the performance of ORACLE and the NO EXPOSURE settings are better than ours. Hence when the noisy annotations are less, using fully exposed label space with some temperature parameter helps the overall performance. For fine-grained labels, we perform better than the no-exposure setting for most categories.

**Performance vs #oracle annotations**. In Fig. 7, we plot the label-wise difference in performances where the brightness of the color bars vary based on the # oracle annotations. For the ORACLE and NO EXPOSURE settings, we observe that our approach works better for labels which have lesser # number of annotations. As the number of annotations increase, we are similar to the baseline settings.

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

Figure 7: MS COCO Label-wise Performance (Sorted by the AP score). Increasing color brightness of the bars indicate increasing **# oracle annotations**.

# A   CIFAR Knowledge Graph

(a) NATURAL

(b) THINGS

Figure 8: CIFAR100 Knowledge Graph

 **B   MS-COCO Knowledge Graph**

(a) NATURAL

(b) MAN-MADE

(a) INDOOR

(b) OUTDOOR

Figure 9: MS-COCO Knowledge Graph