[Reviews · NeurIPS 2020]

Review 1

Summary and Contributions: This paper investigates strategies for learning from partially annotated datasets. In this setting, multiple labels are associated with an image and annotations are provided for some subset of the full list of labels providing both positives and negatives supervised with a BCE loss. The authors compare some straightforward baselines such as treating all un-annotated labels as negatives, and propose their own strategy to soften this loss. A temperature term that governs the penalty associated with an un-annotated label is calculated as a function of label- and image-level distances: - for the label-level distance, the median feature vector of all images associated with each label is computed (using pretrained ImageNet features), and these median vectors are compared between labels - the image-level distance is computed by looking at the nearest neighbor images in ImageNet feature space and whether any of the neighbors has a positive/negative assignment for an un-annotated label The minimum of all these distances for both positive/negative labels is used to calculate the temperature for a given un-annotated label. Evaluation is done on a number of large-scale datasets (COCO, OpenImages) to show that using this soft supervision results in consistently better performance, this can be shown as a fraction of labels provided at training time.

Strengths: - this is an important problem, especially in the context of what is likely to be a common trend of getting partial annotations in large scale dataset collection as seen in Open Images. - results are reported across many different datasets backing up that the soft loss outperforms simpler existing strategies. The results in Fig 3 show that this method allows for much more effective use of small proportions of provided labels - it should be fairly straightforward to compute these temperature terms ahead of time for a given dataset and leverage them throughout training - overall the writing and motivation of the paper is solid

Weaknesses: - the method itself is pretty straightforward, but I wonder a bit at some choices. In particular, so much effort goes into computing many different distances (image-level/label-level and positive/negative), but almost all of this information is discarded because only the min value across all of these is used to compute the final temperature term. It just seems at odds with a "soft" penalty to add so many "hard" operations to throw away other potentially relevant information. Why not do some weighted combination? Were any other alternatives explored? Seems quite possible that one type of distance would tend to dominate, is that the case in practice? - the section describing the core of the method (L193-214) could be much clearer. I had to go through it several times to follow what was going on, and there is not much in the way of high level descriptions of what is taking place and why. - to me it feels like there's a bit of a jump from fairly naive baselines to the somewhat complicated method for computing temperature proposed in this work. Simply adding a constant temperature term for the "label smoothing" baseline already does fairly well, would be interesting to think about what some other straightforward techniques would achieve the desired effect here. For example, it's fairly common in semi-supervised learning to introduce an entropy penalty so that the network makes more "decisive" predictions for unlabeled examples, seems like this would be an idea that could be easily incorporated.

Correctness: Don't see any issues concerning correctness.

Clarity: Besides my concern raised in the weaknesses section, I found the rest of the writing clear and accessible.

Relation to Prior Work: The authors list quite a number of related works, and make good discussions/comparisons, though it is surprising that more existing work wouldn't serve as baselines for the experiments in this paper.

Reproducibility: Yes

Additional Feedback: ### Update post-rebuttal ### The authors addressed a number of my questions in the rebuttal, and as I already had a positive impression of the paper, I maintain my recommendation for acceptance. ####################### - there were a few hyperparameters described in the work, particularly the number of nearest neighbors and the beta/gamma values in the termperature calculation. How were values decided for these hyperparams? Is performance sensitive to the choices for these hyperparams? - since all of the distances are computed on ImageNet features it seems that this would impose a strong bias on what classes are more or less similar. It would be really interesting to see results from bootstrapping and iterating (though understandably beyond the scope of this paper) perhaps initializing with embeddings from the LS baseline to start - is it expensive to go through the whole dataset and compute these temperature terms? - color differences are somewhat difficult to make out in Fig 5, I wonder how the figure would look if instead the x-axis were sorted by most -> least annotations and the trend maybe can be seen a bit clearer, since I think that's the main point of the figure is to see that the biggest boost are to the labels with fewer annotations? Also, the description in L302 I think contradicts the description in the figure caption?


Review 2

Summary and Contributions: This paper introduces a new approach to train a deep ConvNets on image partially annotated with image-level labels. Learning with partial annotations is interesting because it can reduce the cost to build a dataset. This approach exploits relationships among images and labels to define the confidence of the pseudo negative label. This confidence, which is implemented as a temperature parameter, is estimated based on the distance between some labels and images. The proposed approach is validated on several datasets including COCO, COCO panoptic segmentation, OpenImage and LVIS.

Strengths: + The problem of learning deep ConvNets with partial annotated images is interesting because it can reduce the cost to build a dataset. + The authors evaluated their model on several datasets including COCO, COCO panoptic segmentation, OpenImage and LVIS. + The proposed model is simple and gives good performances on several datasets. + Each missing label is considered as a negative label but the authors introduce a temperature parameter to control the confidence to each pseudo negative label. The authors use 2 similarity measures to estimate this temperature parameter: one based on label relationships and one based on image relationships. + The model is evaluated for several proportions of missing labels which is interesting to show the behaviour of the model in different settings. + The authors compared several training approaches (no exposure, weighted no exposure, full exposure and soft exposure) and showed that the soft exposure gives better performances for large proportions of missing labels.

Weaknesses: - The technical contribution seems limited. Using a temperature parameter is not novel and is widely used in a lot of settings. Similarly, computing distances between images and labels was already done but this combination seems novel. - The authors should give more information about the k-NN. For example, they should explain what feature (normalization or other preprocessings) is used. They should explain how they choose k and how sensible is the model to this hyperparameter. - For d_L distances, I wonder what happens when P(x) or N(x) is empty? - How is the category label \psi(c) computed?

Correctness: I did not see flaws in the proposed method.

Clarity: The paper is clear and well written.

Relation to Prior Work: The authors discussed the connections to some recent and/or important works but it is not possible to discuss all the methods because the literature in that topic is huge.

Reproducibility: No

Additional Feedback: The results on the Multi-label CIFAR dataset are not very convincing because it is not a real multi-label dataset but a dataset with hierarchical labels. I think the authors should not show these results in the paper. The authors addressed my concerns so I recommend to accept the paper


Review 3

Summary and Contributions: The paper introduces a method for learning from partially annotated datasets. It is based on comparing the similarity with label/image representations. The model is validated on benchmark data.

Strengths: -impressive experimental results -nice idea and good description of related methods in section 3

Weaknesses: -I observe some inconsistency between results reported in the paper and the results shown in [14]. Are both settings the same? If not, why the authors follow the same setting? If yes, what is the reason? -the objective involves there hyperparameters. The authors state that they were set manually (end of page 5). How to justify this selection? The authors should verify and discuss the sensitivity on their selection. Maybe it is better to select them using validation set? -I think that the authors missed a few papers which also consider the problem of partial labels in classification e.g. https://openaccess.thecvf.com/content_ECCV_2018/papers/Hong-Min_Chu_Deep_Generative_Models_ECCV_2018_paper.pdf

Correctness: yes

Clarity: yes

Relation to Prior Work: There are many more papers which consider the problem of partial labels e.g. https://openaccess.thecvf.com/content_ECCV_2018/papers/Hong-Min_Chu_Deep_Generative_Models_ECCV_2018_paper.pdf

Reproducibility: Yes

Additional Feedback: -it would be interesting to verify how the method performs when only a single label is provided for each image in training -the method uses pretrained network for computing similarities. What to do if we do not have access to such network, e.g. in less explored domains such as cheminformatics? -----------------AFTER REBUTTAL------------------------- I am satisfied with the authors answers and I generally am for accepting the paper. However, I must agree that stronger baselines could be used for comparison. The authors should browse relevant literature.


Review 4

Summary and Contributions: This submission studies how to train on partially annotated image classification datasets. For example the Open Images dataset is partially annotated in the sense that human labelers might say that Apple, Banana are in the image and Pineapples are not in the image, but not say anything about whether a Person is in the image. To train with this kind of partially labeled data, the authors look at “no exposure” (where classes that are not explicitly annotated are simply ignored), “full exposure” (where classes that are not annotated are treated as negatives), and “soft exposure” in which the sigmoid cross entropy loss is label-smoothed by varying amounts depending on similarity of a particular class to one of the positive or negative explicitly provided classes. Authors show some results on simulated settings where some labels are randomly removed from a dataset as well as on the Open Images benchmark.

Strengths: The main result of this paper is that the proposed soft exposure loss outperforms the vanilla baselines of no-exposure of full-exposure on the Open Images benchmark.

Weaknesses: I don’t recommend accepting this paper as it is currently difficult to really know how the proposed method performs against other related work. There are two main reasons: + First, it is hard to compare to other works on metrics --- for example, the authors convert the detection problem (e.g. on COCO and Open Images) to a classification problem (but still evaluate using mean AP (which I am guessing is measured differently from the standard detection mean AP), so it is difficult to say whether the results are strong or not compared to previous work. + Second there is no comparison against related ways of dealing with partial labels (e.g. using graph convolutional neural nets) which the authors mention in the Related Work section. Overall, the experimental results need to be strengthened significantly.

Correctness: Yes.

Clarity: The paper is reasonably well written.

Relation to Prior Work: Conceptually the discussion of prior work is there in the paper, but as noted above, there is no empirical comparison so it is hard to know where this method stands in relation to prior work from a practical perspective.

Reproducibility: Yes

Additional Feedback:

[Author Response · NeurIPS 2020]

We would like to thank the reviewers for their detailed comments and feedback. All new experiment results have conducted on the MS COCO detection dataset (Sec. 4.1.1).

**R1,4.** PRIOR WORK. The state of the art approach for the partially annotated multi-label classification task is [14]. There are two contributions of [14]. One of them is the nWE baseline, which our approach outperforms. The other contribution is using GNN. But it barely has any improvement. In our settings we do see a similar trend of achieving $< 0.2\%$ improvement in mAP. From a high level, instead of modeling label relations directly from the data, we use priors in terms of distance of class embeddings. Moreover, we exploit image similarities as well in this approach.

**R1,2,3.** SENSITIVITY TO HYPER-PARAMETERS. We selected the values of $\beta = 5$ and $\gamma = 0$ based on the validation set. The mAP is within 1% of the reported performance (on average) for $\gamma \in (0, .1]$. The drop in performance can be as large as 5% with $\gamma \in (0, 1]$. For,$\beta \in \{1, 2, 5, 10, 20, 50, 100\}$, avg. mAP on the validation set was within 2% of the best performance at $\beta = 5$. For values of $5 < k \le 30$, the SEI performance increases by 5%, but it's improvement on the SE model is $< 0.15\%$. $k < 5$, reduces the performance of SEI and SE models and brings them closer to NE and SEL models respectively.

**R1.** DESIGN CHOICES OF SE MODELING. The main motivation of the paper is to use image-image and label-label relationships to capture more supervisory signal from the unsupervised un-annotated labels. We implemented this via temperature modeling. Exploring better modeling choices is a work in progress. Thank you for suggesting the entropy based modeling. Regarding the "hard" minimum operations, we also experimented with "softer" operations in this rebuttal. Instead of taking the minimum, we take the median operation in Eq. [4] and [5] among the top 5 neighbors. We see an improvement of $\sim 0.7\%$ for label-label relationships using this approach.
While both label and image based relationships improve the performance, we do observe that the label based distances dominate the image based. This is because the number of labels considered for the image based distances is significantly lesser than the label based distances. While additional 72.7 labels are considered for the label based, the number of labels being considered for image based is $\sim 5.5$ @10% partially annotated data. We will perform an in-depth analysis of the effect of this discrepancy on our approach.
INITIALIZE EMBEDDINGS ON LS BASELINE. This improved our performance by upto **1.5%** mAP.
DISTANCE COMPUTATION COST. It takes <1 epoch training time ($\sim 15$ min. on a single V100 GPU) and it's done once.
PAPER IMPROVEMENTS. Thank you for the feedback. We will improve the writing of the core section as well as the visualization of Fig. 5 in the camera ready draft.

**R2.** FEATURE PRE-PROCESSING FOR DISTANCE COMPUTATIONS. We process the features in the same way as [72], where we use the 2048-dim feature vector and do L2 normalization on them. For k-NN, we use these features to compute the neighbors. For $\psi(c)$, we take a median of these representations across all images where, $c$ occurs. We had also experimented with mean, but found median to have better performance.
$d_L$ DISTANCES WHEN $P(x), N(x) = \phi$. Implementation-wise, we ignore such labels when this happens. However, when combined with $d_I$, the overall distance value defaults to 1 based on Eq. 5.
VALIDITY OF RESULTS ON MULTI-LABEL CIFAR DATASET. As rightly pointed out, CIFAR is indeed a single label dataset and the multiple labels is created because of the hierarchy of the knowledge graph. The purpose of this dataset is to explore the effectiveness of this approach when there is a single object visually present in the image. However, we experiment with other multi-label datasets such as MS COCO detection and panoptic segmentation, and real-world partially annotated multi-label datasets such as OpenImages and LVIS.

**R3.** INCONSISTENCY OF RESULTS WITH [14]. Compared to [14], the main difference is that [14] uses an older split of MS COCO training and validation set (which was taken from an older paper), which is not considered standard in the object detection and multi-label classification literature anymore. We used the training setup of [66] for our experiments. The oracle (with 100% labels) results match that of [66]. We had ran our approach using the setting mentioned in [14], and can conclude the same trend as observed here. We will add these results in our final draft.
SINGLE LABEL PERFORMANCE. Our SE model improves the best performing FE baseline by **13.5%** in mAP.
ABSENCE OF PRE-TRAINED NETWORKS. This is a great question. In a trivial way, we can compute similarities after every "few" epochs. Initially, we can simply use the LS modeling. But we can investigate this issue further and explore meta learning or more sophisticated approaches.
MISSING REFERENCE. Thank you for the missing reference. We will add it in our final draft.

**R4.** CLASSIFICATION VS DETECTION TASKS. Multi-label classification is a well-established task and MS COCO along with OpenImages are considered standard benchmarks for this task. Detection is another task, which along with image-level labels, also require bounding box annotations for localization.

[Meta-Review · NeurIPS 2020]

This paper received reviews from 4 reviewers. The reviewers appreciated the solid, clear exploration of partial annotations, an important problem. The proposed methods are sensible and have positive simplicity. The reviewers engaged in further discussion after reading the authors' rebuttal. Overall, while additional experimentation (e.g. against GCNs) is possible, the current manuscript already provides a substantial amount of empirical work. The demonstration of the effectiveness of these simple methods outperforming more advanced baselines is valuable to the community, and the paper should spur continued interest in this important partial annotation problem space.